# LIME: Link-based user-item Interaction Modeling with decoupled xor attention for Efficient test time scaling

## Abstract

Scaling large recommendation systems requires advancing three major frontiers: processing longer user histories, expanding candidate sets, and increasing model capacity. While promising, transformers' computational cost scales quadratically with the user sequence length and linearly with the number of candidates. This trade-off makes it prohibitively expensive to expand candidate sets or increase sequence length at inference, despite the significant performance improvements.

We introduce **LIME**, a novel architecture that resolves this trade-off. Through two key innovations, LIME fundamentally reduces computational complexity. First, low-rank "link embeddings" enable pre-computation of attention weights by decoupling user and candidate interactions, making the inference cost nearly independent of candidate set size. Second, a linear attention mechanism, **LIME-XOR**, reduces the complexity with respect to user sequence length from quadratic ($O(N^2)$) to linear ($O(N)$).

Experiments on public and industrial datasets show LIME achieves near-parity with state-of-the-art transformers but with a $10\times$ inference speedup on large candidate sets or long sequence lengths. When tested on a major recommendation platform, LIME improved user engagement while maintaining minimal inference costs with respect to candidate set size and user history length, establishing a new paradigm for efficient and expressive recommendation systems.

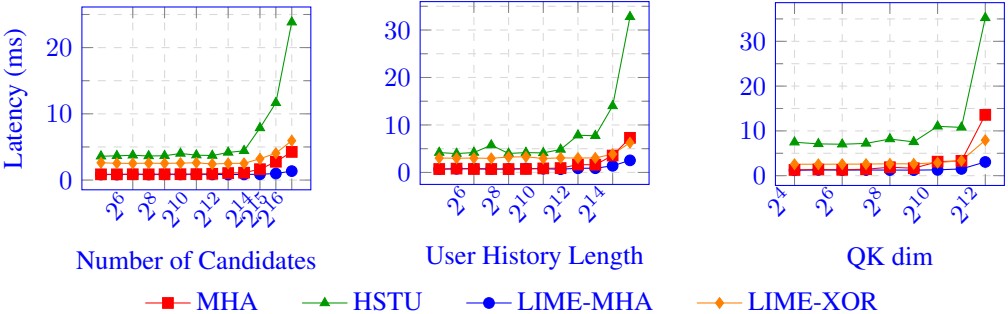

Figure 1: Overall latency analysis across different model parameters. Both LIME models scale well with history length and number of candidates to rank whereas skyline model latencies explode.

## 1 Introduction

Modern recommendation systems operate at a massive scale, facing the challenge of ranking millions of candidate items within strict real-time latency constraints. To succeed, ranking models must navigate a fundamental trade-off between computational efficiency and predictive accuracy. This

---

*Joint first authors.
†Corresponding author.

has led to two dominant but conflicting architectural paradigms. On one end of the spectrum is the two-tower model (Covington et al., 2016a), which achieves unparalleled inference speed by encoding users and items into separate, independent representations. This separation enables the use of efficient approximate nearest-neighbor search (Shrivastava & Li, 2014), making real-time recommendations feasible at scale.

On the other end are powerful cross-attention Transformer models like SASRec (Kang & McAuley, 2018b), which deliver state-of-the-art accuracy by encoding users' long interaction history (UIH) via multi-layer self attention and explicitly modeling deep, contextual interactions between that and each candidate item. Although this approach provides rich expressiveness, it comes with a significant computational cost: the self-attention over long sequences and the cross-attention between the full user sequence and each candidate become major performance bottlenecks. This architectural dilemma is becoming increasingly acute. The push for higher quality recommendations now demands scaling along three axes simultaneously: accommodating vast candidate sets, processing longer user histories, and deploying models of increasing complexity. These demands are fundamentally at odds, as existing efficient models lack expressiveness, while expressive models lack efficiency. Hybrid approaches (Li et al., 2022) offer only incremental improvements.

To fundamentally solve this problem and enable scaling along all three axes, we propose a new architectural blueprint: the Link-based User-Item Interaction Modeling for Efficient inference (LIME) framework. LIME is designed from the ground up to achieve the modeling power of a full cross-attention system while operating within the strict efficiency budget of a two-tower model. Its central innovation is a globally learned intermediate "link embedding" sequence that acts as a bridge between the long user history and candidate items. This design decouples the user and item representations during online inference, making scoring independent of history length by pre-computing the most expensive attention components offline. Furthermore, it enables the introduction of a new low rank attention mechanism to reduce user interaction history (UIH) self-attention complexity from quadratic to linear. By resolving these two primary computational bottlenecks, LIME provides a comprehensive solution to the expressiveness-versus-efficiency challenge, enabling deep interaction modeling at minimal latency (Figure 1).

Our primary contributions are as follows:

**The Link Embedding Mechanism:** We introduce a link embedding sequence that effectively approximates full cross-attention. This mechanism allows the expensive Query-Key attention weight, $\phi(\tilde{Q}\tilde{K}^{\top})$, to be pre-computed and cached offline, enabling cross-attention-like expressiveness with the efficiency of Two-Tower modeling during online inference.

**XOR Attention Masking:** To overcome the quadratic time complexity of Transformers with respect to user history length, we propose an XOR attention mask that factorizes the full self-attention matrix into a bidirectional linear attention between the link embeddings and the user history sequence.

**State-of-the-Art Performance and Impact:** We demonstrate through extensive experiments that LIME achieves performance competitive with computationally intensive ranking models like HSTU (Zhai et al., 2024) with 10x lower latency. Deployed in production, LIME has yielded up to 38% source rate[1] gain on a major platform serving billions of users.

## 2 RELATED WORK

LIME addresses the long-standing trade-off between model expressiveness and inference efficiency in large-scale ranking. We situate our contributions in the context of two primary research areas: efficient ranking architectures and innovations in sequence modeling.

### 2.1 EFFICIENT RANKING ARCHITECTURES

The design of ranking models is dominated by a conflict between efficiency and interaction depth. On one end of the spectrum, two-tower models (Covington et al., 2016b; Yi et al., 2019) achieve unparalleled efficiency. By encoding users and items into separate embedding spaces, they enable fast candidate retrieval using Approximate Nearest Neighbor (ANN) search. However, this sepa-

---

[1]This refers to the percentage of positively engaged items attributable to the ranking model using LIME.

ration prevents deep, feature-level interactions, limiting model expressiveness. On the other end, cross-attention models like DIN (Zhou et al., 2018) enable rich, target-aware interactions by dynamically attending to user history for each candidate, but their per-item computational cost makes them prohibitive for ranking large candidate sets.

Several approaches have sought to bridge this gap. Hybrid models add shallow interaction layers on top of a two-tower base (Li et al., 2022), though these often provide only marginal improvements. LIME offers a more fundamental solution. It introduces a novel bridge built upon a fixed set of global, learnable parameters, which we term link embeddings. These link embeddings act as an intermediary, allowing LIME to capture the rich dynamics of cross-attention while retaining the architectural efficiency of a two-tower model. This design, where a static, user-independent key space (raw links) retrieves information from a dynamic, personalized value space (personalized links), is conceptually similar to key-value memory networks (Miller et al., 2016) and the retrieval stage of retrieval-augmented models, enabling massive pre-computation. This approach directly tackles the sequential nature of user history and the need for scalable target-item interaction, striking a new balance between efficiency and expressiveness.

## 2.2 EFFICIENT ATTENTION FOR LONG SEQUENCE MODELING

The Transformer's quadratic complexity ($O(N^2)$) for self-attention remains a fundamental bottleneck for modeling long sequences. This challenge has driven the recent advances in Large Language Models (LLMs), moving beyond early approximations such as kernelization (Choromanski et al., 2021) or low-rank projections (Wang et al., 2020). More recent breakthroughs include new model classes—State Space Models (e.g., Mamba (Gu & Dao, 2023))—and hardware-aware techniques such as Lightning Attention (Dao et al., 2024) that directly optimize attention computation.

Within recommender systems, progress has often focused on adapting NLP-inspired attention mechanisms to click-through rate (CTR) prediction (Zhang et al., 2022; Li et al., 2023; Song et al., 2025) or using pruning to shorten sequences before attention is applied (Pi et al., 2020). While general-purpose mechanisms for handling set-based inputs exist, they often do not align with the specific needs of sequential recommendation. For instance, the Set Transformer (Lee et al., 2019) and its Pooling Multi-Head Attention (PMA) mechanism efficiently summarize a set into a fixed-size representation using learnable seed vectors. However, their primary goal is permutation-invariant summarization, whereas ranking requires target-aware representations that preserve the sequential nature of user histories.

LIME's XOR Attention contributes a distinct, task-specific solution. Rather than being a general-purpose approximation of the self-attention matrix, it is a mechanism co-designed with the LIME architecture. By using link embeddings as intermediaries, LIME structurally eliminates the need for direct history-to-history self-attention at inference time. This design enforces a linear complexity ($O(L \cdot N)$) tailored specifically for the user–item ranking context, representing a novel approach to building efficient and expressive sequence models for recommendations.

## 3 MODEL OVERVIEW

We propose the **L**ink-based user-item **I**nteraction **M**odeling for **E**fficient inference (**LIME**), a novel sequential-modeling architecture tailored for Click-Through Rate (CTR) prediction. LIME is designed to bridge the gap between highly efficient but less expressive two-tower models and powerful but computationally expensive cross-attention architectures. We present its design by progressively building from a simple, efficient baseline to a scalable model with deeper interactions.

To represent a user $U$'s interaction history, we learn embedding tables to generate embeddings for each of the $N(U)$ items the user has interacted with. Each item is characterized by a set of attributes, which can be categorical (e.g., user action, topic id) or continuous (e.g., video length), and we learn embedding tables for each attribute. Continuous features are first transformed into categorical ones via bucketization to index into the embedding tables. For each item, we concatenate the learned embeddings of all its attributes and project them through a Multi-Layer Perceptron (MLP) to obtain a unified representation, $E_j$. The entire user history is then represented as a sequence of these embeddings, $E = \{E_j\}_{j=1}^{N(U)} \in \mathbb{R}^{N(U) \times d}$.

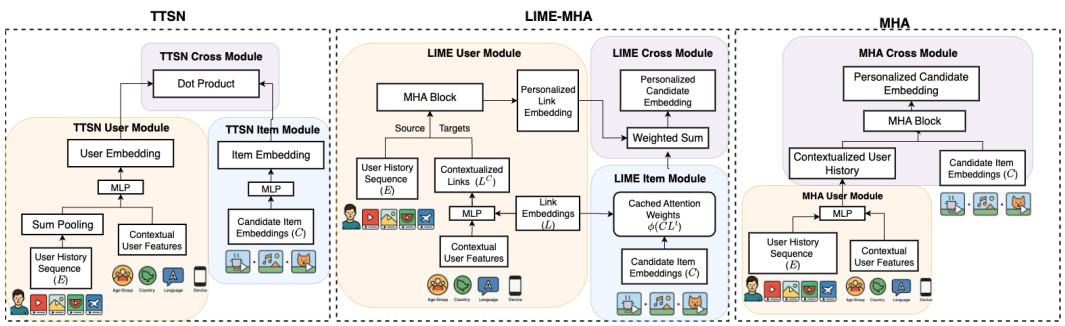

Figure 2: Architecture of TTSN (left), LIME-MHA (middle), and MHA (right). With a lightweight cross module using precomputed attention weights in a decoupled attention framework, LIME-MHA achieves MHA-level expressiveness with similar latency to TTSN.

## 3.1 FROM TWO-TOWERS TO LIME-MHA: BRIDGING EFFICIENCY AND EXPRESSIVENESS

While a two-tower model (TTSN) is highly efficient for scoring millions of items in retrieval, its expressivity is limited, as user-item interaction is confined to a late-stage dot product. At the other extreme, full cross-attention, prototyped by Multi-Head Attention (MHA), allows deep interaction between every candidate item and the entire user history. However, its computational cost, which scales with both candidate count ($M(U)$) and history length ($N(U)$), places the entire burden on the online interaction stage, making it prohibitively expensive.

LIME-MHA bridges this gap by introducing a small, fixed-size set of $\ell \ll N(U)$ auxiliary tokens, or **link embeddings** $L \in \mathbb{R}^{\ell \times d}$. These learned embeddings act as a compact summary of user interests, which are first personalized based on the user's history and then exposed to candidate items. This factorization is achieved via two MHA stages, effectively reducing the complexity from $O(M(U) \cdot N(U))$ to $O(\ell(M(U) + N(U)))$.

Multi-Head Attention (MHA), introduced by Vaswani et al. (Vaswani et al., 2017), is a function that maps $q$ query vectors $Q \in \mathbb{R}^{q \times d}$ to outputs using $n$ key and value vectors $K, V \in \mathbb{R}^{n \times d}$ as:

$$\mathrm{MHA}(Q, K, V; M; \theta) = \left(\phi(\tilde{Q}\tilde{K}^{\top}) \odot M\right)\tilde{V} \tag{1}$$

where $\tilde{Q} = QW_Q$, $\tilde{K} = KW_K$, $\tilde{V} = VW_V$, with learnable parameters $\theta = \{W_Q, W_K, W_V \in \mathbb{R}^{d \times d}\}$. Here, $\phi$ is an activation function (e.g., scaled Softmax or SiLU) and $M \in \{0, 1\}^{q \times n}$ is a binary mask. For intance, $J[i, j] := \equiv 1$ is the trivially all-1 mask pattern. $(A \odot B)_{ij} := A_{ij}B_{ij}$ stands for Hadamard product (elementwise multiplication).

The LIME-MHA architecture operates in two steps:

**1. User-Side Link Personalization.** The globally shared link embeddings $L$ are first contextualized with user features $E^C$ (e.g., location, device type) via an MLP:

$$L^C = \mathrm{MLP}(L \oplus E^C) \tag{2}$$

These contextualized links are then personalized by attending to the user's full interaction history $E$ using a single MHA layer, producing personalized link embeddings $L^P$:

$$L^P = \mathrm{MHA}(L^C, E, E; J; \theta) \tag{3}$$

**2. Candidate-Side Decoupled Interaction.** A key innovation of LIME is how candidate (target) embeddings $T$ interact with the personalized links. Instead of a standard MHA where keys and values both come from $L^P$, we use the raw, user-independent link embeddings $L$ as keys:

$$O = \mathrm{MHA}(T, L, L^P; J; \theta) = \phi(TL^t)L^P \tag{4}$$

This seemingly small change has a profound impact on efficiency. The attention weight matrix, $\phi(TL^t) \in \mathbb{R}^{M(U) \times \ell}$, is now independent of the user. It can be pre-computed offline for all items in the corpus and cached. At inference time, this expensive matrix multiplication is replaced by a simple lookup, and the interaction reduces to a lightweight weighted sum of the personalized link embeddings $L^P$. This makes the serving latency per candidate effectively constant, $O(1)$, rather than scaling with history length.

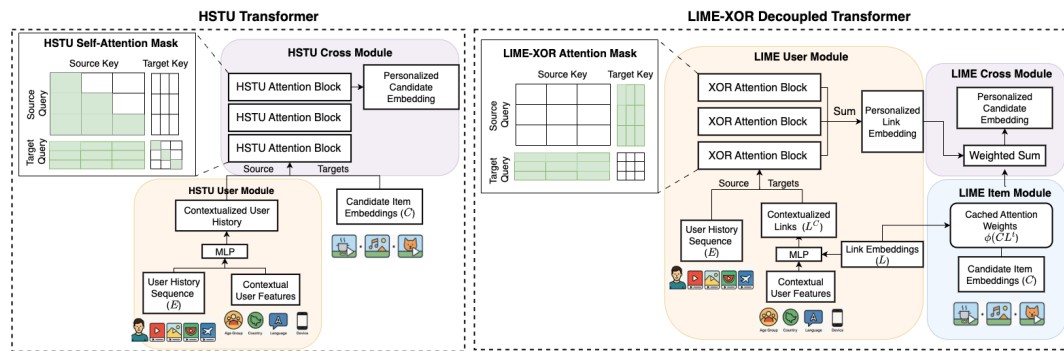

Figure 3: Architecture of HSTU Transformer (left) and LIME-XOR Decoupled Transformer (right) with a visual comparison of our proposed XOR attention mask against the standard HSTU causal self-attention mask.

## 3.2 FROM LIME-MHA TO LIME-XOR: SCALING TO DEEPER INTERACTIONS

To enhance LIME's expressiveness, we can deepen the user-side module by stacking multiple interaction layers, akin to the architecture of a multi-layer Transformer. This computation is performed only once per user request, so its complexity does not impact the per-candidate scoring latency. A state-of-the-art approach for this would be to adapt a powerful sequential model block, such as the Hierarchical Sequential Transducer Unit (HSTU) (Zhai et al., 2024), which can be conceptually summarized as follows (see Appendix A for full notations):

$$\text{HSTUBlock}(X; \theta) = \text{GatedMLP}(\text{MHA}(X, X, X; M_{\text{causal}}; \theta); \eta) \tag{5}$$

where $X = E \oplus L^C$ is the concatenation of the user history and contextualized link embeddings. However, the standard HSTU attention uses causal self-attention mask, $M_{\text{causal}}[i, j] := i \geq j$, where every token in the user history attends to all preceding tokens (self-attention) and every candidate attends to the entire user history (cross-attention), results in a computational complexity of $O(N(U)^2 + N(U) \cdot M(U))$. This creates a major bottleneck for users with long interaction sequences or large candidate sets to rank.

To overcome this, we introduce XOR Attention (**XORA**), a novel attention kernel designed to replace the standard self-attention mechanism within the user-side module.

$$\text{XORA}(X, X, X; \theta) = \text{MHA}(X, X, X; M_{\text{xor}}; \theta) \tag{6}$$

$$= \text{MHA}(E, L^C, L^C; J; \theta) \oplus \text{MHA}(L^C, E, E; J; \theta) \tag{7}$$

where $M_{\text{xor}}[i, j] := 1_{i \in [0, |E|)} \wedge 1_{j \in [0, |E|)}$ is the exclusive-or mask pattern that ensures the source and target embeddings attend to one another only. As depicted in Figure 3, the XOR mask structurally eliminates the expensive history-to-history ($E \leftrightarrow E$) interactions. Instead, it facilitates an efficient, two-way, block-wise attention (7): the link embeddings attend to the user history ($L^C \rightarrow E$), and crucially, the user history (source) embeddings now also attend back to the link embeddings ($E \rightarrow L^C$).

This modification provides two main advantages. First, it reduces the computational complexity from quadratic to linear, $O(\ell \cdot N(U))$, making deep, multi-layer processing of long histories feasible. Second, it enriches the model's expressivity by enabling the user history representation to be modulated by the global context of the link embeddings from the very first layer.

This leads to our advanced variant, **LIME-XOR**. In this model, we define an **XOR-Layer** by replacing the causal mask in (5) with our efficient XOR mask, $M_{\text{XOR}}$ in (6). The personalized links are then computed by stacking and summing the outputs of $n$ such layers:

$$L^P = \sum_{j=1}^{n} \text{GatedMLP}\left(\text{XORA}\left(E \oplus L^C, E \oplus L^C, E \oplus L^C; \theta\right); \eta_j\right) \tag{8}$$

This entire deep computation occurs on the user side, preserving the efficient, per-candidate scoring mechanism and LIME's overall scalability. Note that by using contextual links $L^C$ as targets instead

Figure 4: LIME Inference as part of the entire recommendation system

of candidate embeddings $T$, the process remains fully decoupled from individual candidates during this stage.

### 3.3 LIME'S ARCHITECTURAL ADVANTAGES FOR EFFICIENT INFERENCE

The LIME framework is designed from the ground up to resolve the conflict between model expressiveness and inference latency. Its core efficiency stems from a strategic decoupling of user-side and item-side computations, bridged by the link embeddings. This design enables a multi-stage inference pipeline that maximizes offline pre-computation and minimizes online work, making it highly scalable for real-world recommendation systems.

To facilitate this efficiency, we employ a technique called **decoupled attention**, used in the final candidate interaction stage (Equation 4). Instead of using the user-specific personalized links ($L^P$) for both keys and values, we use the raw, user-independent link embeddings ($L$) as keys and the personalized links ($L^P$) as values. This asymmetric structure ensures that the expensive Query-Key dot product, $\phi(CL^t)$, depends only on item-side information (candidate embeddings $C$ and raw links $L$). Consequently, this attention weight matrix can be pre-computed for all items in the corpus and stored in an efficient key-value store or index, such as FAISS (Johnson et al., 2017), effectively creating a cache of attention weights.

This architectural choice allows us to structure the entire inference process into three distinct stages, as illustrated in Figure 4.

**Offline Item-Side Pre-computation.** This stage is performed offline whenever the model or item catalog is updated, eliminating redundant computation during online serving. Standard item features are processed by an item tower to produce item embeddings. The decoupled attention weights ($\phi(CL^\top)$) between all candidate item embeddings ($C$) and the raw link embeddings ($L$) are pre-computed and cached. This transforms the most intensive part of cross-attention into a simple lookup.

**Online User-Side Computation.** This stage runs once per user request and is independent of the number of candidates being scored. The user's context and interaction history are processed by the LIME user module (using either MHA or the multi-layer XOR Transformer) to produce the personalized link embeddings ($L^P$), as described in Equations (2)–(8). Crucially, this user-side computation can be executed in parallel with the candidate retrieval process. In a production environment, its latency is therefore largely masked, making even a deep, multi-layer Transformer on the user side feasible.

**Lightweight User-Item Interaction.** This final stage is executed online for each candidate but is extremely lightweight. For each candidate, the pre-computed attention weights are retrieved from the QK Cache. These weights are used to perform a simple weighted sum over the personalized link embeddings ($L^P$) to generate the final LIME embedding. This embedding is then passed to a shallow interaction network for scoring.

By structuring inference this way, LIME achieves significant computational savings. Compared to a full cross-attention model like HSTU, which has a complexity of $\mathcal{O}(N(U) \cdot (N(U) + M(U)))$, LIME's complexity is reduced to $\mathcal{O}(\ell \cdot (N(U) + M(U)))$, where $\ell \ll N(U), M(U)$. More im-

portantly, Model serving is reduced to a near-constant time operation with respect to the number of candidates, making LIME highly suitable for latency-sensitive ranking deployments with large candidate sets and long user histories.

## 4 EXPERIMENTAL RESULTS

### 4.1 BASELINES

We test multiple baseline models ranging from the simplest sum-pooling of user history embeddings to other sequence compression techniques listed below.

**TTSN** (Two-tower sparse network): Depicted in Figure 2, this baseline applies sum-pooling of all the user history embeddings instead of target attention with candidate embeddings.

### 4.2 SKYLINES

We cannot launch naive target attention (candidate against user history) directly for long user histories and large numbers of candidates to rank. But such powerful models can serve as our skyline goal on performance.

**MHA Skyline** computes a single layer of target attention between candidate items and user history items.

**HSTU Skyline** computes 3 layers of causal self attention and cross attention at each layer, between the candidate items and user history items.

### 4.3 LIME VARIANTS

To thoroughly evaluate our proposed architecture, we conducted experiments with two primary variants of LIME that differ specifically in the sophistication of the link personalization module.

**LIME-MHA**: This is the fundamental implementation of our model, as described in Equation (3). It employs a standard attention mechanism to generate a personalized representation for each link by pooling information from the user's historical item embeddings.

**LIME-XOR**: This advanced variant, detailed in Equation (8), incorporates 3-layers of HSTU between the user history and link embeddings for deeper contextualization with our proposed efficient XOR-style masking.

**LIME-XOR+Window**: This variant extends the LIME-XOR attention mask by adding a sliding-window pattern over the user history, where each token attends to its $\ell$ neighbors on both sides (similar to Longformer (Beltagy et al., 2020)). The attention remains linear: each token attends to $2\ell$ tokens—$\ell$ link tokens and $\ell$ local neighbors. XOR attention captures global interactions between user history tokens, while the windowed attention captures local interactions.

To ensure a fair and controlled comparison, all other architectural components were held constant across all models (see details in Appendix C).

### 4.4 ACCURACY

We present the normalized entropy metrics as well as (session) AUC metrics for all the variants and baselines/skylines. On the in-house industrial datasets, we measure accuracy on two tasks: video completion (VC) and watch time (WT).

Normalized Entropy (NE) (He et al., 2014) is defined for binary classification task as

$$\text{NE}(\{(p_i, \ell_i)\}_{i=1}^n) := \frac{\sum_{i=1}^n \ell_i \log p_i + (1 - \ell_i) \log(1 - p_i)}{\log(\sum_{i=1}^n \ell_i / n)}.$$

Note that similar to logloss or binary cross entropy, lower NE means better accuracy. Usually $0.1\%$ drop in NE will lead to online metric improvement.

Figure 5: Offline, online, and public dataset experimental results.

### (a) Industrial Results

| Model | Offline | | Online A/B | |
|---|---|---|---|---|
| | VC NE | WT AUC | VC | WT |
| TTSN | – | – | – | – |
| LIME-MHA | -0.72% | +0.46% | +28.6% | +22.1% |
| MHA Sky. | **-0.73%** | **+0.53%** | N/A | N/A |
| LIME-XOR | -1.04% | +0.76% | **+37.9%** | **+28.6%** |
| LIME-XOR+Window | **-1.11%** | **+0.80%** | N/A | N/A |
| HSTU Sky. | -1.06% | +0.77% | N/A | N/A |

MHA and HSTU Skylines cannot be tested in online setting due to high serving latency from the large number of candidates.

### (b) Public Dataset Results

| Model | KuaiRand-1K | Taobao-Ad |
|---|---|---|
| | Click AUC | AUC |
| TTSN | 0.7389 (+0%) | 0.6452 (+0%) |
| DIN | 0.7404 (+0.20%) | 0.6468 (+0.25%) |
| SASRec | 0.7419 (+0.41%) | 0.6462 (+0.15%) |
| Trunc. MHA | 0.7351 (-0.51%) | 0.6456 (+0.06%) |
| LREA | 0.7408 (+0.26%) | 0.6447(-0.08%) |
| LIME-MHA | **0.7433** (+0.60%) | **0.6465** (+0.20%) |
| MHA Sky. | 0.7428 (+0.53%) | 0.6464 (+0.19%) |
| LIME-XOR | 0.7448 (+0.80%) | 0.6467 (+0.23%) |
| LIME-XOR+Window | **0.7453** (+0.87%) | 0.6468 (+0.25%) |
| HSTU Sky. | 0.7444 (+0.74%) | **0.6475** (+0.36%) |

## 4.5 Industrial Experiments Setup

For the industrial dataset, we take 3 days of logged data for training, and 6 hours for evaluation. Compared to the public datasets, this is a larger scale dataset with longer user history sequences. As shown in Table 5, both LIME variants significantly outperform the TTSN baseline.

LIME-MHA nearly matches the MHA skyline, while the multi-layer LIME-XOR closes the gap further, achieving performance competitive with the much more complex HSTU skyline across all tasks, even with a 32x sequence compression rate. The 1.04% VC NE improvement of LIME-XOR over the TTSN baseline is a significant gain, which translated to a +37.9% VC and +28.6% WT increase from LIME-ranked candidates during online A/B experiments (Table 5).

## 4.6 Public Experiments Setup

We also benchmark on public datasets, namely Taobao Ads (Lyu et al., 2020) and KuaiRand-1K dataset (Gao et al., 2022). Taobao-Ads contains 25 million interactions with a maximum sequence length of 50 whereas KuaiRand-1K contains 12 million interactions with a maximum sequence length of 256. To unify data processing and evaluation framework, we leaned heavily on FuxiCTR, a comprehensive sequential recommendation model benchmark platform (Zhu et al., 2021).

On both public datasets, LIME-MHA matches or outperforms the respective skyline with significant improvements over the sum-pooling baseline. The improvement is largest in the KuaiRand-1K dataset as it contains longer sequences (length 256). On KuaiRand-1K, LIME-XOR performs comparably to the HSTU skyline but slightly worse on Taobao-Ad, though still outperforming LIME-MHA models. This is likely due to shorter sequence lengths in Taobao-Ad, leading to noisier results.

We also benchmark existing sequence-compression methods, including Truncated MHA—which applies the MHA skyline only to the most recent $\ell$ interacted items—and Linformer-style LREA (Song et al., 2025) with a low rank of $\ell$. At the same low rank, LIME outperforms both compression techniques. Additionally we benchmark skyline models SASRec (Kang & McAuley, 2018a) (a state-of-the-art transformer) and DIN (Deep Interest Network) (Zhou et al., 2018) to validate our LIME variants and skylines are indeed improvements over existing state-of-the-art models.

## 4.7 Ablation Study & Scaling Laws

We ablate various components of LIME-XOR to demonstrate the effectiveness of its design by considering the following variants:

**LIME-XOR w/ Link Pooling**: Instead of decoupled attention with candidate embeddings, we apply sum pooling to the personalized link embeddings.

**LIME-XOR w/ Dot Product**: Instead of decoupled attention with candidate embeddings, we pass the $\ell$ dot products between candidate embedding and personalized link embeddings to the final prediction MLP.

Figure 6: Ablation Study and comparison to linear sparse attention variants on KuaiRand-1K dataset.

(a) Ablation Study

| Model Variant | AUC |
|---|---|
| LIME-XOR | **0.7448** |
| LIME-XOR w/ Link Pooling | 0.7400 |
| LIME-XOR w/ Dot Product | 0.7404 |
| LIME-MLP | 0.7407 |
| LIME-XOR–MLP Hybrid | 0.7442 |

(b) Comparison to SOTA Linear Attention Variants

| Model Variant | AUC | Complexity |
|---|---|---|
| HSTU Skyline | 0.7444 | $O(N(U)^2 + N(U) \cdot M(U))$ |
| Linear Transformer | 0.7423 | $O(N(U) \cdot M(U))$ |
| Longformer | 0.7443 | $O(N(U) \cdot M(U))$ |
| LREA (Linformer) | 0.7408 | $O(N(U) \cdot M(U))$ |
| LIME-XOR | 0.7448 | $O(N(U)\ell + M(U)\ell)$ |
| LIME-XOR+Window | **0.7453** | $O(N(U)\ell + M(U)\ell)$ |

**LIME-MLP**: Instead of leveraging attention mechanisms, we directly learn MLPs to generate personalized low-rank embeddings (after padding the user history to fixed length) and cached weights.

**LIME-XOR-MLP**: We preserve the LIME-XOR user module with multi-layer XOR attention and replace the LIME item module cached attention weights with weights learned from an MLP on the candidate item embeddings.

The ablation results in Figure 6(a) demonstrate that both multi-layer XOR attention from the LIME user module and the decoupled attention mechanism in the LIME item module are critical to strong performance. The attention mechanisms cannot easily be replaced by MLP, sum pooling, or dot products in either user or item module especially due to variable user history sequence lengths.

For both NLP and CTR prediction tasks, transformer-based models (e.g., LLMs, HSTU) have shown improved performance as compute scales. On our large-scale ranking dataset, the decoupled transformer LIME-XOR similarly achieves substantial gains when scaling sequence length, link count, and model depth. Detailed results are provided in Appendix G.

## 4.8 SPARSE LINEAR ATTENTION VARIANTS

We note that several alternate sparse linear attention mechanisms exist such as Linear Transformer (Katharopoulos et al., 2020), Longformer (Beltagy et al., 2020), and Linformer-style LREA (Wang et al., 2020; Song et al., 2025). These can be integrated into the skyline transformer to reduce the quadratic self-attention cost ($O(N(U)^2)$) to linear $O(N(U)\ell)$ for low-rank $\ell$, though they will still suffer from $O(N(U) \cdot M(U))$ target attention cost. We can also integrate these linear attention variants within our proposed decoupled LIME framework to also optimize the target attention cost (e.g. LIME-XOR+Window).

From Figure 6(b), both Linear Transformer and Linformer-based methods have severe performance degradation but sliding-window based Longformer performs comparably to the HSTU skyline. However, it is still prohibitively expensive to serve (see Appendix J for latency analysis) due to the cross-attention cost. Yet, when we integrate sliding window attention into LIME-XOR we see improved performance with minimal latency, demonstrating the benefits of the LIME-XOR decoupled attention framework.

## 4.9 INFERENCE SPEED

LIME is highly scalable for ranking large candidate sets, an advantage for pre-ranking and retrieval. Figure 1 demonstrates that while MHA and HSTU latency grows significantly with more candidates or longer user histories, LIME's latency remains nearly constant (see Appendix F for a detailed comparison). This robustness makes it suitable for settings with very long sequences (e.g.,>30k items). Furthermore, the user backbone computation can be parallelized with candidate retrieval, masking most of its latency in production and yielding even greater savings than shown in these benchmarks.

## 5 ANALYSIS AND DISCUSSIONS

LIME effectively projects the high-dimensional user-user and user-candidate interaction spaces into low-rank subspaces, acting as a surrogate attention mechanism. This decomposition can be viewed as a structured approximation of the full attention matrix, analogous to techniques in sparse/dense low-rank compression (Figure 8).

To this end, we compute the singular value decomposition (SVD) of the self-attention and cross-attention matrices in a trained HSTU Transformer skyline model averaged across layers and heads over 256 sequence-candidate pairs where each sequence has length 1024 and we rank 1024 candidates against the sequence. In Figure 7, the results clearly demonstrate a long-tailed pattern where the largest 32 singular values (denoted by the vertical black line) capture more than 90% of the information in both self and cross-attention matrices. We also compute the SVD of the raw link embeddings $L$ and personalized link embeddings $L^P$ of a trained LIME-XOR model and analyze the spectral distribution. The singular values of $L$ and $L^P$ demonstrate strong separation amongst the link embeddings with nearly full rank. This analysis indicates that 32 links are able to capture the majority of the information in the self-attention and cross-attention matrices with minimal redundancy for sequence lengths of 1024.

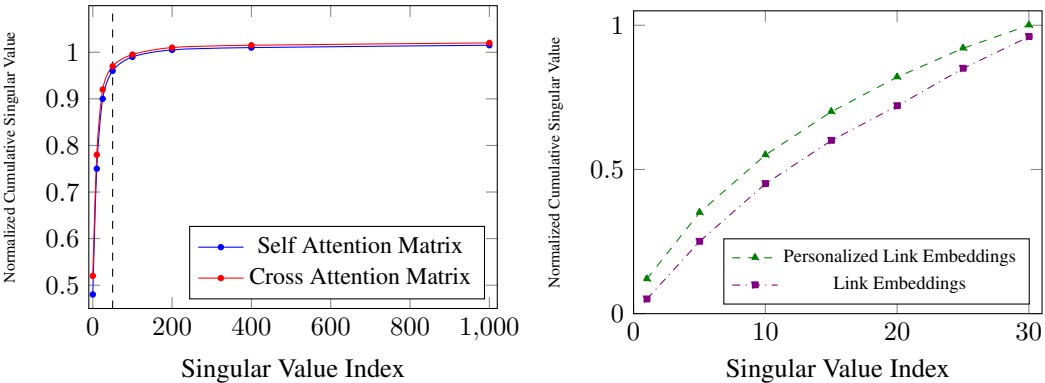

Figure 7: Left: cumulative singular value of self/cross attention matrices in a pretrained transformer model. Right: cumulative singular value of the raw/personalized link embeddings.

## 6 LIMITATIONS AND FUTURE WORK

Despite LIME's strong empirical performance, several limitations warrant discussion. First, LIME's effectiveness relies on the low-rank approximation assumption; domains with highly fragmented user interests may require higher-rank representations. Second, pre-computing QK caches requires periodic updates for evolving item catalogs, which makes it better suited for early stage ranking.

We are actively exploring the following directions: (1) building a larger pool of link embeddings and dynamically selecting link subsets based on user-side features or context, enabling personalized link allocation across different user segments, (2) extensions to multi-modal recommendation and cross-domain transfer learning to address the cold start user and item problem.

## 7 CONCLUSION

We proposed LIME, a framework that resolves the efficiency-expressiveness trade-off in large-scale recommenders by using link embeddings as a low-rank approximation for target attention or more general transformer style sequence encoders. Experiments on public and industrial datasets show LIME matches the accuracy of state-of-the-art models like HSTU while drastically reducing inference latency, a result confirmed by significant online A/B test improvements. Our analysis confirmed that LIME's learned link embeddings effectively capture user interests, validating the low-rank hypothesis. By decoupling the user representation from target items, LIME offers a practical and powerful solution for building high-performance, scalable recommender systems.

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

## A SUMMARY OF NOTATIONS

We summarize the key notations used in the paper in Table 1.

| Symbol | Description |
|---|---|
| $U$ | User |
| $N(U)$ | Number of items in user $U$'s interaction history |
| $M(U)$ | Number of candidate items for user $U$ |
| $E_j$ | Embedding representation of item $j$ |
| $E = \{E_j\}_{j=1}^{N(U)}$ | Sequence of user history embeddings, $E \in \mathbb{R}^{N(U) \times d}$ |
| $E^C$ | User context features (e.g., location, device type) |
| $T$ | Candidate (target) item embeddings |
| $d$ | Embedding dimension |
| $L \in \mathbb{R}^{\ell \times d}$ | Link embeddings (auxiliary tokens), $\ell \ll N(U)$ |
| $L^C$ | Contextualized link embeddings (personalized with user features) |
| $L^P$ | Personalized link embeddings (after attention to user history) |
| $\mathrm{MHA}(Q, K, V; M; \theta)$ | Multi-Head Attention function |
| $W_Q, W_K, W_V$ | Learnable projection matrices for queries, keys, values ($\in \mathbb{R}^{d \times d}$) |
| $\theta, \eta$ | Set of learnable parameters |
| $\phi$ | Activation function (e.g., scaled Softmax, SiLU) |
| $O$ | Final LIME output embedding after candidate-side MHA |
| $M$ | Binary mask for attention, $M \in \{0, 1\}^{q \times n}$ |
| $J$ | All-ones mask pattern, $J[i, j] \equiv 1$ |
| $M_{\mathrm{causal}}$ | Causal self-attention mask, $M_{\mathrm{causal}}[i, j] := i \geq j$ |
| $M_{\mathrm{xor}}$ | XOR attention mask, $M_{\mathrm{xor}}[i, j] := 1_{i \in [0,|E|)} \wedge 1_{j \in [0,|E|)}$ |
| $\odot$ | Hadamard (elementwise) product |
| $\oplus$ | Concatenation operator |
| GatedMLP | MLP block with parameters $\eta$ to learn a gated element-wise product |
| $\mathrm{HSTUBlock}(X; \theta)$ | Hierarchical Sequential Transducer Unit block with parameters $theta$ |
| XORA | XOR Attention Block |
| $n$ | Number of stacked XOR layers |

Table 1: Notation Table

## B  RANKING MODEL ARCHITECTURE COMPARISONS

Outlined in Table 2, two-tower based models have the highest scalability in both candidate set size and user history length but suffer from extremely limited (e.g. dot product) late interaction. Conversely, transformer-based models have deep user-item interactions through full self-attention (amongst user history) and cross-attention (between candidates and user history) at every layer. However, transformer-based models suffer from high latency and low scalability on both user history length and candidate set size.

SIM (Pi et al., 2020) and TWIN (Chang et al., 2023) improve scalability over Transformer-based methods through a two-stage approach. The general-search unit (GSU) first searches for the top-$K$ relevant user history interactions for a particular candidate and exact-search unit (ESU) performs a full cross-attention against the retrieved items. While the latency is reduced compared to Transformer, we also sacrifice some model expressivity when we completely remove user history self-attention and the GSU runtime complexity is $O(MN)$ which can still be inefficient.

On the other hand, LIME is able to model deep user-item interactions (through the decoupled Transformer framework introduced in Section 3) and achieves high scalability to both candidate set size and user history length when $L \ll N$.

## C  EXPERIMENT IMPLEMENTATION DETAILS

### C.1  DATASET PREPARATION

For the Taobao Ad dataset, we leverage the preprocessed version provided by the FuxiCTR (Zhu et al., 2021) sequential modeling platform. For the KuaiRand dataset (Gao et al., 2022), we pre-

Table 2: Comparison of Ranking Model Architectures where $N$ is the user history length, $M$ is the number of candidates to rank, and $L \ll N$ is the LIME-compressed history size.

| Axis | Transformers (HSTU) | SIM/TWIN | Two-Tower | LIME (Ours) |
|---|---|---|---|---|
| User-Item Interaction | Deep | Medium | Limited late interaction | Deep |
| Latency | High | Medium | Low | Low |
| Complexity | $O(MN + N^2)$ | $O(MN)$ | $O(N)$ | $O(ML + NL)$ |
| Pre-computation | Minimal | Minimal | Item Emb | QK Attention Weights |

process it ourselves by partitioning the first 14 days of interactions for training and last 2 days for testing. We also discarded all items from the train and test set with less than 30 total interactions for more reliable results. For the industrial dataset, we took 3 days of logged data for model training and 6 hours of data for evaluation.

### C.1.1 MODEL HYPERPARAMETERS

For both public datasets and the industrial datasets, we fix model hyperparameters and seeds for all variants for a fair comparison (see Table 3). The industrial dataset is of the largest scale both in terms of the number of interactions and maximum sequence length. Due to the large size of the industrial dataset, we only train for a single epoch in a streaming single-pass setting for the industrial dataset.

| Parameter | KuaiRand | TaoBao Ad | Industrial |
|---|---|---|---|
| Num. of Interactions | $12 \times 10^6$ | $25 \times 10^6$ | $100 \times 10^9$ |
| Learning Rate | $1 \times 10^{-4}$ | $1 \times 10^{-3}$ | $4 \times 10^{-4}$ |
| Batch Size | 1024 | 8192 | 1024 |
| Number of Heads | 4 | 4 | 4 |
| Epochs | 2 | 10 | 1 |
| Embedding Dimension | 32 | 32 | 256 |
| Number of Links | 16 | 8 | 32 |
| Max Sequence Length | 256 | 50 | 1024 |
| Interaction MLP | [512, 128, 64] | [512, 256, 128] | [96] |

Table 3: Model Hyperparameters for KuaiRand, TaoBao Ad, and Industrial datasets

### C.2 MODEL DESIGN

Several additional design choices were crucial to stabilize training and ensure generalization:

- **Normalization:** We apply Layer normalization before each linear projection in the QKV projections. Without normalization, the embeddings can drift toward high magnitudes, which can collapse attention weights.

- **Link Initialization:** The raw link embeddings $\ell_i$ are initialized with samples from a standard normal distribution. This encourages diversity and allows the model to discover interpretable interest clusters during training.

- **Attention Function:** In single-layer MHA experiments we use scaled Softmax which performs the best and in multi-layer transformer experiments we leverage SiLU as the attention activation $\phi$.

- **Training Objective:** We use a standard cross-entropy loss with multi-task objectives per request. Other loss terms (e.g., contrastive objectives or auxiliary disentanglement losses) may be explored in future work.

## D LOW RANK EXPRESSIONS FOR SELF-ATTENTION AND CROSS-ATTENTION

LIME effectively projects the high-dimensional user-user and user-candidate interaction spaces into low-rank subspaces, acting as a surrogate attention mechanism:

$$\text{MHA}(T, E) \approx \text{MHA}(T, L) \cdot \text{MHA}(L, E),$$
$$\text{MHA}(E, E) \approx \text{MHA}(E, L) \cdot \text{MHA}(L, E).$$

This decomposition can be viewed as a structured approximation of the full attention matrix, analogous to techniques in sparse/dense low-rank compression (e.g. Performer, Linformer, SVDNet).

## E    INFORMATION BOTTLENECK PERSPECTIVE

From an information-theoretic viewpoint, LIME compresses the user history $E$ into a set of link embeddings $L$, which are optimized to retain relevance to the user's behavior (via personalized attention pooling) and retain discriminative power for candidate ($T$) ranking.

This fits into the Information Bottleneck (IB) principle:

$$\min_L \mathcal{I}(L; E) - \beta\mathcal{I}(L; T)$$

where $\mathcal{I}$ denotes mutual information. That is, we retain only those aspects of the user history that are useful for predicting interaction with candidates.

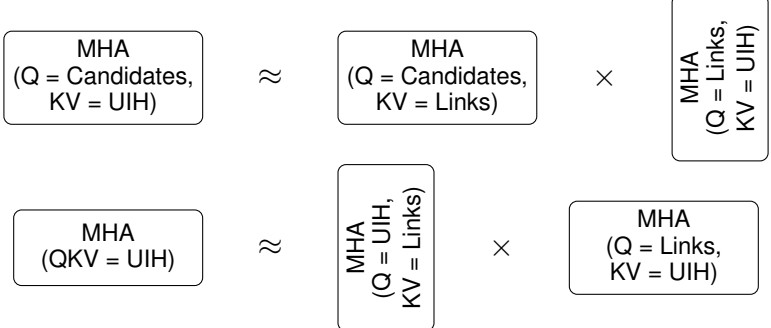

Figure 8: Top: Cross-attention as a low-rank product. Bottom: Self-attention as two low-rank cross-attentions. UIH stands for user interaction history.

## F    INFERENCE SPEED COMPARISON

Depicted in Figure 1 and Table 4, we observe that LIME-MHA and LIME-XOR have significantly lower latencies under large candidate sets to rank, long user histories, and high QK dimensions (due to the attention weight cacheing).

## G    SCALING LAW FOR LIME-XOR DECOUPLED TRANSFORMER

Transformers (such as HSTU) have demonstrated strong scaling law for large-scale recommendation systems, mirroring scaling laws from LLMs from NLP. For our decoupled transformer, LIME-XOR, we test scaling across three axes on the large-scale industrial dataset: sequence length, number of links, and number of layers.

From Table 5, we observe that scaling sequence length is very effective in improving both VC NE and WT AUC. However, it is only effective under the presence of sufficient links. Scaling sequence length from 2k to 4k under 32 links is neutral but similar scaling with 256 links demonstrates significant NE and AUC wins. Moreover, for shorter sequences (e.g. 1k) scaling the number of links seems to reach an inflection point faster than scaling links for longer sequences.

From Table 6, we observe that scaling number of layers demonstrates significant NE and AUC improvements which helps demonstrate that LIME-XOR can also achieve scaling law up to the limits we were able to test.

Table 4: Latency (in seconds) comparison for different models under varying conditions. The columns correspond to the models: LIME-MHA, MHA Skyline, HSTU Skyline, and LIME-XOR. All numerical values are truncated to two decimal places.

(a) vs. Number of Candidates (FlashAttention V2)

| # Cand. | LIME-MHA | MHA Sky. | HSTU Sky. | LIME-XOR |
|---|---|---|---|---|
| 16 | 0.84 | 0.87 | 3.60 | 2.62 |
| 32 | 0.83 | 0.87 | 3.66 | 2.53 |
| 64 | 0.88 | 0.88 | 3.74 | 2.52 |
| 128 | 0.84 | 0.87 | 3.64 | 2.55 |
| 256 | 0.87 | 0.88 | 3.67 | 2.52 |
| 512 | 0.85 | 0.88 | 4.01 | 2.56 |
| 1024 | 0.87 | 0.89 | 3.76 | 2.61 |
| 2048 | 0.85 | 0.95 | 3.70 | 2.43 |
| 4096 | 0.83 | 1.06 | 4.17 | 2.51 |
| 8192 | 0.86 | 1.11 | 4.42 | 2.53 |
| 16384 | 0.87 | 1.64 | 7.86 | 3.21 |
| 32768 | 0.99 | 2.76 | 11.66 | 4.06 |
| 65536 | 1.35 | 4.25 | 23.83 | 5.95 |

(b) vs. User History Length (FlashAttention V2)

| Hist. Len. | LIME-MHA | MHA Sky. | HSTU Sky. | LIME-XOR |
|---|---|---|---|---|
| 16 | 0.68 | 0.72 | 4.18 | 3.02 |
| 32 | 0.76 | 0.80 | 3.97 | 3.00 |
| 64 | 0.83 | 0.72 | 4.19 | 3.01 |
| 128 | 0.71 | 0.71 | 5.75 | 2.97 |
| 256 | 0.72 | 0.70 | 4.00 | 3.25 |
| 512 | 0.81 | 0.73 | 4.17 | 3.29 |
| 1024 | 0.77 | 0.85 | 4.11 | 2.94 |
| 2048 | 0.68 | 0.91 | 4.83 | 3.05 |
| 4096 | 0.87 | 1.63 | 7.83 | 3.04 |
| 8192 | 0.88 | 1.62 | 7.72 | 3.06 |
| 16384 | 1.39 | 3.56 | 14.00 | 3.84 |
| 32768 | 2.54 | 7.29 | 32.78 | 6.25 |

(c) vs. QK Dimension (PyTorch)

| QK dim | LIME-MHA | MHA Sky. | HSTU Sky. | LIME-XOR |
|---|---|---|---|---|
| 16 | 1.22 | 1.28 | 7.46 | 2.56 |
| 32 | 1.28 | 1.38 | 7.10 | 2.55 |
| 64 | 1.25 | 1.31 | 7.02 | 2.57 |
| 128 | 1.26 | 1.44 | 7.21 | 2.59 |
| 256 | 1.26 | 1.88 | 8.18 | 2.65 |
| 512 | 1.24 | 1.66 | 7.54 | 2.64 |
| 1024 | 1.30 | 3.13 | 11.02 | 2.90 |
| 2048 | 1.51 | 3.31 | 10.78 | 3.36 |
| 4096 | 3.07 | 13.57 | 35.23 | 7.93 |

|      | 32      | 64      | 128     | 256     |
|------|---------|---------|---------|---------|
| 1k   | 0%      | -0.04%  | -0.19%  | -0.20%  |
| 2k   | -0.16%  | -0.19%  | -0.34%  | -0.39%  |
| 4k   | -0.11%  | -0.36%  | -0.36%  | **-0.55%** |

(a) VC NE % improvement across sequence lengths and link counts

|      | 32      | 64      | 128     | 256     |
|------|---------|---------|---------|---------|
| 1k   | 0%      | +0.01%  | +0.08%  | +0.11%  |
| 2k   | +0.03%  | +0.05%  | +0.18%  | +0.18%  |
| 4k   | +0.03%  | +0.13%  | +0.17%  | **+0.27%** |

(b) WT AUC across sequence lengths and link counts

Table 5: Performance across varying sequence lengths (1k, 2k, 4k) and link counts (32, 64, 128, 256). Each cell reports metric value and relative change from the baseline (1k, 32 links).

| Number of Layers | VC NE % Improvement | WT AUC % Improvement |
|------------------|---------------------|----------------------|
| 3                | 0%                  | 0%                   |
| 6                | -0.25%              | +0.12%               |
| 9                | -0.34%              | +0.18%               |
| 12               | **-0.38%**          | **+0.19%**           |

Table 6: VC NE and WT AUC for increasing model depth on fixed 1k sequence length with 32 links.

# H DERIVATIONS FOR XOR ATTENTION KERNEL

## H.1 FORWARD PASS

Let the output of the XOR-attention be $O = (O[S], O[T])$, where $S, T$ stand for source and target respectively. In the context of LIME, source is the user history item embeddings, while target is the link embeddings. Similarly define $Q[S], Q[T], K[S], K[T], V[S], V[T]$ to be the source and target portion of the query, key, value embedding sequences.

$$O[T] = \phi(Q[T]K[S]^\top)V[S]$$
$$O[S] = \phi(Q[S]K[T]^\top)V[T]$$

## H.2 BACKWARD PASS

Let $dO[T]$ denote an infinitesimally small change in $O[T]$, also known as the its differential. Similarly define $dO[S], dV[S], dV[T], dQ[S], dQ[T], dK[S], dK[T]$.

Trivially we have

$$\frac{\partial O[T]}{\partial Q[S]} = \frac{\partial O[T]}{\partial K[T]} = \frac{\partial O[T]}{\partial V[T]} = 0$$
$$\frac{\partial O[S]}{\partial Q[T]} = \frac{\partial O[S]}{\partial K[S]} = \frac{\partial O[S]}{\partial V[S]} = 0$$

The total differential of the loss function is given by

$$dL = \text{Tr}\left(\left(\frac{\partial L}{\partial O[S]}\right)^\top dO[S]\right) + \text{Tr}\left(\left(\frac{\partial L}{\partial O[T]}\right)^\top dO[T]\right).$$

Since $dO[S] = \phi(Q[S]K[T]^\top)dV[T]$ and $dO[T] = \phi(Q[T]K[S]^\top)dV[S]$,

$$dL_V = \text{Tr}\left(\left(\frac{\partial L}{\partial O[S]}\right)^\top \phi(Q[S]K[T]^\top)dV[T]\right) + \text{Tr}\left(\left(\frac{\partial L}{\partial O[T]}\right)^\top \phi(Q[T]K[S]^\top)dV[S]\right).$$

By comparing coefficient with the matrix chain rule $dL_{V[T]} = \text{Tr}\left(\left(\frac{\partial L}{\partial V[T]}\right)^\top dV[T]\right)$, we see that

$$\frac{\partial L}{\partial V[T]} = \phi(K[T]Q[S]^\top)\frac{\partial L}{\partial O[S]}.$$

Similarly

$$\frac{\partial L}{\partial V[S]} = \phi(K[S]Q[T]^\top)\frac{\partial L}{\partial O[T]}.$$

Next for derivatives with respect to $Q$, we have

$$dL_{Q[S]} = \text{Tr}\left(\left(\frac{\partial L}{\partial O[S]}\right)^\top \phi'(Q[S]K[T]^\top)V[T]K[T]^\top dQ[S]\right)$$

$$dL_{Q[T]} = \text{Tr}\left(\left(\frac{\partial L}{\partial O[T]}\right)^\top \phi'(Q[T]K[S]^\top)V[S]K[S]^\top dQ[T]\right).$$

Hence

$$\frac{\partial L}{\partial Q[S]} = K[T]V[T]^\top \phi'(K[T]Q[S]^\top)\frac{\partial L}{\partial O[S]}$$

$$\frac{\partial L}{\partial Q[T]} = K[S]V[S]^\top \phi'(K[S]Q[T]^\top)\frac{\partial L}{\partial O[T]}.$$

Finally from

$$dL_{K[T]} = \text{Tr}\left(\left(\frac{\partial L}{\partial O[S]}\right)^\top \phi'(Q[S]K[T]^\top)V[T]Q[S]^\top dK[T]\right)$$

$$dL_{K[S]} = \text{Tr}\left(\left(\frac{\partial L}{\partial O[T]}\right)^\top \phi'(Q[T]K[S]^\top)V[S]Q[T]^\top dK[S]\right),$$

we get

$$\frac{\partial L}{\partial K[S]} = Q[T]V[S]^\top \phi'(K[S]Q[T]^\top)\frac{\partial L}{\partial O[T]}$$

$$\frac{\partial L}{\partial K[T]} = Q[S]V[T]^\top \phi'(K[T]Q[S]^\top)\frac{\partial L}{\partial O[S]}$$

## I TRITON PSEUDOCODE

---
**Algorithm 1: XOR Mask and Denominator Computation**

---
**Require:** Query index $i$, key index $j$, number of sources $n_s$
**Ensure:** Binary mask $m$, normalization denominator $d$
  1: $\text{is\_src}_q \leftarrow (i < n_s)$
  2: $\text{is\_src}_k \leftarrow (j < n_s)$
  3: $m \leftarrow \text{is\_src}_q \oplus \text{is\_src}_k$                 ▷ Attend iff exactly one is source
  4: $d \leftarrow \text{is\_src}_q \,?\, n_t : n_s$              ▷ Normalize by opposite partition size
  5: **return** $m, d$

---

---
Algorithm 2: Forward Pass with Block Range Selection

---

**Require:** $\mathbf{Q}, \mathbf{K}, \mathbf{V} \in \mathbb{R}^{n \times d}$, number of sources $n_s$
**Ensure:** Output $\mathbf{O} \in \mathbb{R}^{n \times d}$

1: **for** $q_{\text{start}} \leftarrow 0$ to $n$ step $B_M$ **do**                    $\triangleright$ Parallel over query blocks
2:     $\mathbf{Q}_b \leftarrow \mathbf{Q}[q_{\text{start}} : q_{\text{start}} + B_M, :]$                    $\triangleright$ Load to SRAM
3:     $\text{acc} \leftarrow \mathbf{0}_{B_M \times d}$
4:
5:     **// Block range selection (critical optimization)**
6:     **if** $q_{\text{start}} + B_M \leq n_s$ **then**                    $\triangleright$ Pure source queries
7:         $k_{\text{lo}}, k_{\text{hi}} \leftarrow n_s, n$                    $\triangleright$ Load target keys only
8:     **else if** $q_{\text{start}} \geq n_s$ **then**                    $\triangleright$ Pure target queries
9:         $k_{\text{lo}}, k_{\text{hi}} \leftarrow 0, n_s$                    $\triangleright$ Load source keys only
10:    **else**                    $\triangleright$ Boundary case
11:        $k_{\text{lo}}, k_{\text{hi}} \leftarrow 0, n$
12:    **end if**
13:
14:    **for** $k_{\text{start}} \leftarrow k_{\text{lo}}$ to $k_{\text{hi}}$ step $B_N$ **do**                    $\triangleright$ Over selected K,V blocks
15:        $\mathbf{K}_b \leftarrow \mathbf{K}[:, k_{\text{start}} : k_{\text{start}} + B_N]$                    $\triangleright$ Load K,V to SRAM
16:        $\mathbf{V}_b \leftarrow \mathbf{V}[k_{\text{start}} : k_{\text{start}} + B_N, :]$
17:        $\mathbf{S} \leftarrow \mathbf{Q}_b \mathbf{K}_b$                    $\triangleright$ Compute scores (tensor cores)
18:
19:        **for** $i \leftarrow 0$ to $B_M$ **do**                    $\triangleright$ Mask & normalize in registers
20:            **for** $j \leftarrow 0$ to $B_N$ **do**
21:                $m, d \leftarrow \text{XORMASK}(q_{\text{start}} + i, k_{\text{start}} + j, n_s)$
22:                $\mathbf{S}[i, j] \leftarrow m \; ? \; \text{SiLU}(\mathbf{S}[i, j])/d : 0$
23:            **end for**
24:        **end for**
25:
26:        $\text{acc} \leftarrow \text{acc} + \mathbf{S}\mathbf{V}_b$                    $\triangleright$ Accumulate (tensor cores)
27:    **end for**
28:
29:    $\mathbf{O}[q_{\text{start}} : q_{\text{start}} + B_M, :] \leftarrow \text{acc}$                    $\triangleright$ Write to HBM
30: **end for**
31: **return** $\mathbf{O}$

---

---

Algorithm 3: Backward Pass (Transposed Access Pattern)

---

**Require:** $\mathbf{dO}, \mathbf{Q}, \mathbf{K}, \mathbf{V}$, number of sources $n_s$
**Ensure:** Gradients $\mathbf{dQ}, \mathbf{dK}, \mathbf{dV}$
 1: $\mathbf{dQ}, \mathbf{dK}, \mathbf{dV} \leftarrow \mathbf{0}$
 2: **for** $k_{\text{start}} \leftarrow 0$ to $n$ step $B_N$ **do**                           ▷ Parallel over K,V blocks
 3:     $\mathbf{K}_b \leftarrow \mathbf{K}[:, k_{\text{start}} : k_{\text{start}} + B_N]$               ▷ Load K,V to SRAM (resident)
 4:     $\mathbf{V}_b \leftarrow \mathbf{V}[k_{\text{start}} : k_{\text{start}} + B_N, :]$
 5:     $\mathbf{dK}_{\text{acc}}, \mathbf{dV}_{\text{acc}} \leftarrow \mathbf{0}_{d \times B_N}, \mathbf{0}_{B_N \times d}$
 6:
 7:     **if** $k_{\text{start}} + B_N \leq n_s$ **then**                      ▷ Source K,V block
 8:         $q_{\text{lo}}, q_{\text{hi}} \leftarrow n_s, n$                      ▷ Process target queries only
 9:     **else if** $k_{\text{start}} \geq n_s$ **then**                     ▷ Target K,V block
10:         $q_{\text{lo}}, q_{\text{hi}} \leftarrow 0, n_s$                      ▷ Process source queries only
11:     **else**                                       ▷ Boundary case
12:         $q_{\text{lo}}, q_{\text{hi}} \leftarrow 0, n$
13:     **end if**
14:
15:     **for** $q_{\text{start}} \leftarrow q_{\text{lo}}$ to $q_{\text{hi}}$ step $B_M$ **do**             ▷ Over selected Q blocks
16:         $\mathbf{Q}_b \leftarrow \mathbf{Q}[q_{\text{start}} : q_{\text{start}} + B_M, :]$
17:         $\mathbf{dO}_b \leftarrow \mathbf{dO}[q_{\text{start}} : q_{\text{start}} + B_M, :]$
18:         $\mathbf{S} \leftarrow \mathbf{Q}_b \mathbf{K}_b$                  ▷ Recompute forward (activation remat)
19:
20:         **for** $i \leftarrow 0$ to $B_M, j \leftarrow 0$ to $B_N$ **do**
21:             $m, d \leftarrow \text{XORMASK}(q_{\text{start}} + i, k_{\text{start}} + j, n_s)$
22:             **if** $m$ **then**
23:                 $s_{ij} \leftarrow \text{SiLU}(\mathbf{S}[i, j])/d$
24:                 $\mathbf{S}[i, j] \leftarrow s_{ij} \cdot (1 - s_{ij}) \cdot (1 + \mathbf{S}[i, j])/d$     ▷ SiLU gradient
25:             **else**
26:                 $\mathbf{S}[i, j] \leftarrow 0$
27:             **end if**
28:         **end for**
29:
30:         $\mathbf{dV}_{\text{acc}} \leftarrow \mathbf{dV}_{\text{acc}} + \mathbf{S}^\top \mathbf{dO}_b$           ▷ Accumulate (tensor cores)
31:         $\mathbf{dK}_{\text{acc}} \leftarrow \mathbf{dK}_{\text{acc}} + (\mathbf{S}^\top \mathbf{Q}_b)^\top$
32:         $\text{ATOMICADD}(\mathbf{dQ}[q_{\text{start}} : q_{\text{start}} + B_M, :], (\mathbf{dO}_b \mathbf{V}_b^\top)\mathbf{S}^\top)$
33:     **end for**
34:
35:     $\mathbf{dK}[:, k_{\text{start}} : k_{\text{start}} + B_N] \leftarrow \mathbf{dK}_{\text{acc}}$          ▷ Write gradients
36:     $\mathbf{dV}[k_{\text{start}} : k_{\text{start}} + B_N, :] \leftarrow \mathbf{dV}_{\text{acc}}$
37: **end for**
38: **return** $\mathbf{dQ}, \mathbf{dK}, \mathbf{dV}$

---

## J   Linear Attention Performance & Latency Analysis

In Figure 9, we compare various linear attention methods (not decoupled so still more expensive than LIME) to both LIME variants and skyline HSTU. The results demonstrate that LIME variants consistently match (or exceed) HSTU skyline AUC under varying sequence length and QK-dimension settings.

We benchmark all variants using PyTorch's FlexAttention library (Dong et al., 2024) which enables us to flexibly compare various masking strategies, ensuring that unnecessary blocks of computation are skipped, without writing custom Triton logic for each type of linear attention. In Figure 10, we observe that Longformer is less expensive than Skyline HSTU when scaling user history length and number of candidates. However, it is still more expensive than LIME decoupled variants in complexity ($O(NM)$ vs $O(N\ell + M\ell)$). We observe it is more than twice as expensive across all user history lengths and spikes at user histories longer than 4096 and at number of candidates to rank greater than 2048.

Figure 9: AUC Comparison of Sparse Attention Variants with columns reporting on a particular Sequence Length/QK Dimension (e.g. 256/32 means 256 sequence length, 32 QK-dimension).

| Variant | 256/128 | 64/128 | 256/32 | 64/32 | Complexity |
|---|---|---|---|---|---|
| HSTU Skyline | 0.7444 | 0.7300 | 0.7425 | **0.7296** | $O(N^2 + NM)$ |
| Linear Trans. | 0.7423 | 0.7292 | 0.7404 | 0.7278 | $O(NM)$ |
| Longformer | 0.7443 | 0.7300 | 0.7425 | 0.7292 | $O(NM)$ |
| Linformer (LREA) | 0.7408 | 0.7289 | 0.7406 | 0.7275 | $O(NM)$ |
| LIME-XOR | 0.7448 | 0.7301 | **0.7427** | 0.7295 | $O(N\ell + M\ell)$ |
| LIME-XOR+Win. | **0.7453** | **0.7305** | 0.7425 | 0.7292 | $O(N\ell + M\ell)$ |

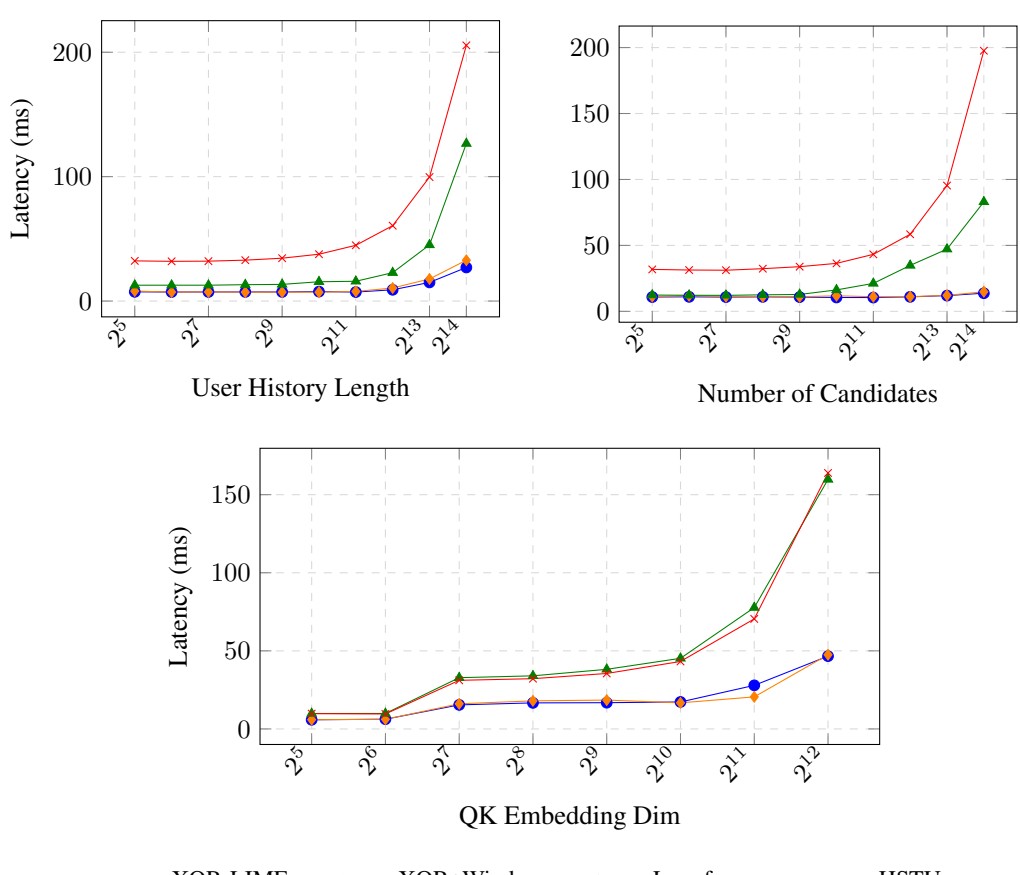

Figure 10: Latency comparison of linear attention to LIME-XOR and HSTU.

## K   LINK CLUSTERING ANALYSIS

For one of the LIME-XOR models trained on the industrial dataset, we pick several links and examine the items that have the largest attention weights with that particular link. In doing so, we find that the clustered items for each link share strong semantic properties (e.g., video topic, length, etc.). Some examples include:

- **Link 2**: Long videos
- **Link 3**: Food & Cooking videos
- **Link 5**: Dog & Animal videos
- **Link 10**: Furniture & DIY videos

- **Link 30**: Sports videos
- **Link 31**: Foreign videos

## L  LIME PERFORMANCE ON COLD-START USERS

Although LIME is not explicitly designed for cold-start users, we observe that it delivers topline improvements for this group in online A/B testing that are comparable to those seen in the overall user cohort. To further understand this empirical behavior, we evaluate our pretrained HSTU skyline and LIME-XOR models on the cold-start subset of KuaiRand-1k. In this evaluation (see Table 7), we vary the maximum number of prior interactions a user may have to qualify as a cold-start user.

| Cold-Start User Threshold | LIME-XOR | HSTU |
|---|---|---|
| $N(U) \leq 4$ | **0.7022** | 0.7016 |
| $N(U) \leq 8$ | **0.7179** | 0.7173 |
| $N(U) \leq 16$ | **0.7284** | 0.7273 |
| $N(U) \leq 32$ | **0.7369** | 0.7356 |

Table 7: LIME-XOR vs HSTU Performance (AUC) on the cold-start user cohort in KuaiRand-1k

One hypothesis for why LIME leads to improved performance for cold-start users is that LIME compresses a user's history into a fixed length embedding (equal to number of links) regardless of whether the user is a power user or cold-start user. This normalization allows the model to extract useful patterns even from sparse histories, reducing variance and improving robustness compared to models like HSTU that depend more directly on the number of prior interactions.

## M  LLM USAGE IN PREPARATION

We used LLM significantly to polish the language of all sections in the paper, including the abstract and appendix sections. The tikz codes in Figure 1, 7, 8, and 10 were generated based on data points we collected ourselves. Most of the tables were also formatted by LLM.

