# OpenReview forum: "LIME: Link-based user-item Interaction Modeling with decoupled xor attention for Efficient test time scaling"
_ICLR.cc/2026/Conference — ICLR 2026 Conference Desk Rejected Submission_

### Official Review · Reviewer_VtZ9 · 2025-10-28

**Soundness:** 2
**Presentation:** 3
**Contribution:** 2
**Rating:** 4
**Confidence:** 3

**Summary:**

This paper presents LIME, which aims to solve the complexity of standard Transformer with self-attention for tackling long user sequence. Theoretically, it reduces the theoretical complexity from $O(N^2)$ to $O(N)$. Empirically, it demonstrates that the speedup happens given long sequence ($N\geq 4096$)

**Strengths:**

The paper provides a figure immediately after abstract to showcase their empirical results.

The paper is written well with fluent narrative.

The paper targets the quadratic complexity of MHA, which is a significant problem especially in Recsys.

Online AB test results are provided.

**Weaknesses:**

According to Figure 1, MHA outperforms the proposed LIME for user sequence lengths $N<4096$, and similarly outperforms LIME-XOR for $N<8192$. This implies that LIME provides limited acceleration benefits within these ranges, which are typical in real-world recommender systems. Since users with longer behavioral sequences (>4096 or 8192) constitute a small fraction, the practical advantage of LIME in terms of acceleration appears limited under realistic conditions.

An ablation study is essential to demonstrate the contribution of individual components in LIME to both acceleration and accuracy. The paper currently lacks such analysis. The authors are encouraged to include an ablation section, comparing LIME against both the standard Transformer and state-of-the-art accelerated Transformer variants, and to report results on model accuracy and computational complexity.

It is of great importance  to add comparisons with more recently developed accelerated attention mechanisms. These comparisons should evaluate both accuracy and efficiency across a comprehensive range of hyperparameters, including user sequence length, number of candidates, and QK dimensionality. The comparisons in the existing literature should also be unified to include these aspects.

The manuscript should include a discussion section addressing the limitations of the proposed approach and outlining potential directions for future research to address these shortcomings.

**Questions:**

In comparison with the baseline MSA, the authors should specify the details of its implementation. It is unclear whether MSA is implemented using the modern FlashAttention mechanism (which could be essential for fairness). The authors are advised to clarify this point and, if FlashAttention is used, to indicate the specific version employed.

---

> ### Author Response · Authors · 2025-11-21
>
> We thank the reviewer for very careful review and highly detailed feedback. We made changes to the paper based on your suggestions, highlighted in blue.  We will address each of your questions and concerns below.
>
> > *According to Figure 1, MHA outperforms the proposed LIME for user sequence lengths N<4096, and similarly outperforms LIME-XOR for N<8192. This implies that LIME provides limited acceleration benefits within these ranges, which are typical in real-world recommender systems. Since users with longer behavioral sequences (>4096 or 8192) constitute a small fraction, the practical advantage of LIME in terms of acceleration appears limited under realistic conditions.*
>
> In our short-form video platform, typical user histories routinely exceed tens of thousands of events. Users often watch hundreds of videos per day due to their extremely short duration, and we also log non-video interactions. As feature logging improves and we aggregate behavior across product surfaces, this sequence length continues to grow, with a long-term target approaching one million. This scale aligns with other work on ultra-long sequences, including TWIN V2 [1], LONGER [2], and PinFM [3].
> Our earlier benchmarks were run in native PyTorch, as the goal at that stage was simply to illustrate LIME’s scalability on long sequences. Those benchmarks also used single-layer Transformers, whereas our updated, production-based results use at least three layers. In addition, the reported sequence lengths reflect the maximum user history, meaning the average length is roughly half of that.
>
> With the updated Triton-based benchmark across all variants, we observe that LIME-XOR consistently outperforms HSTU at every sequence length (see Figure 1). LIME-MHA also begins to surpass standard MHA once the maximum sequence length reaches 256 (average length 128). We additionally removed the logarithmic latency axis from earlier figures to clarify that, although MHA is slightly faster than LIME-MHA at very short sequences, the gap is small and largely attributable to LIME’s two-stage kernel-launch overhead.
> Critically, in production, LIME’s decoupled architecture allows the user module to be precomputed offline or triggered as soon as the user request arrives, effectively hiding its latency. As a result, the user-module cost is eliminated during online inference, making LIME substantially faster than what the paper’s benchmark numbers alone suggest.
>
> The below table provides raw latency data for our updated Triton-based benchmark (visualized in Figure 1 in the paper and Table 4 of Appendix F). Note that while each of these scales only along a single dimension (candidates, history length, QK dimension), in practice we would like to scale along all three of these dimensions.
>
> Table 1: Candidates
>
> | # Cand. | LIME-MHA | MHA Sky. | HSTU Sky. | LIME-XOR |
> |---------|----------|----------|-----------|----------|
> |   256   |   0.87   |   0.88   |   3.67    |   2.52   |
> |   512   |   0.85   |   0.88   |   4.01    |   2.56   |
> |  1024   |   0.87   |   0.89   |   3.76    |   2.61   |
> |  2048   |   0.85   |   0.95   |   3.70    |   2.43   |
> |  4096   |   0.83   |   1.06   |   4.17    |   2.51   |
> |  8192   |   0.86   |   1.11   |   4.42    |   2.53   |
> | 16384   |   0.87   |   1.64   |   7.86    |   3.21   |
> | 32768   |   0.99   |   2.76   |  11.66    |   4.06   |
> | 65536   |   1.35   |   4.25   |  23.83    |   5.95   |
>
> Table 2: History Length
>
> | Hist. Len. | LIME-MHA | MHA Sky. | HSTU Sky. | LIME-XOR |
> |------------|----------|----------|-----------|----------|
> |    256     |   0.72   |   0.70   |   4.00    |   3.25   |
> |    512     |   0.81   |   0.73   |   4.17    |   3.29   |
> |   1024     |   0.77   |   0.85   |   4.11    |   2.94   |
> |   2048     |   0.68   |   0.91   |   4.83    |   3.05   |
> |   4096     |   0.87   |   1.63   |   7.83    |   3.04   |
> |   8192     |   0.88   |   1.62   |   7.72    |   3.06   |
> |  16384     |   1.39   |   3.56   |  14.00    |   3.84   |
> |  32768     |   2.54   |   7.29   |  32.78    |   6.25   |
>
> Table 3: QK Dimension
>
> | QK dim | LIME-MHA | MHA Sky. | HSTU Sky. | LIME-XOR |
> |--------|----------|----------|-----------|----------|
> |   32   |   1.28   |   1.38   |   7.10    |   2.55   |
> |   64   |   1.25   |   1.31   |   7.02    |   2.57   |
> |  128   |   1.26   |   1.44   |   7.21    |   2.59   |
> |  256   |   1.26   |   1.88   |   8.18    |   2.65   |
> |  512   |   1.24   |   1.66   |   7.54    |   2.64   |
> | 1024   |   1.30   |   3.13   |  11.02    |   2.90   |
> | 2048   |   1.51   |   3.31   |  10.78    |   3.36   |
> | 4096   |   3.07   |  13.57   |  35.23    |   7.93   |
>
> [1] TWIN V2: Scaling Ultra-Long User Behavior Sequence Modeling for Enhanced CTR Prediction at Kuaishou
>
> [2] LONGER: Scaling Up Long Sequence Modeling in Industrial Recommenders
>
> [3] PinFM: Foundation Model for User Activity Sequences at a Billion-scale Visual Discovery Platform

---

> ### Author Response · Authors · 2025-11-21
>
> > *An ablation study is essential to demonstrate the contribution of individual components in LIME to both acceleration and accuracy. The paper currently lacks such analysis. The authors are encouraged to include an ablation section, comparing LIME against both the standard Transformer and state-of-the-art accelerated Transformer variants, and to report results on model accuracy and computational complexity.*
>
> That’s a great suggestion. We have added ablation studies (see table 6(a) in section 4.7) for various components of LIME, such as replacing LIME-XOR decoupled attention with
> * sum pooling of the personalized link embeddings
> * dot product between the candidate embedding and link embeddings
> * MLP applied to padded user history embedding sequences on user side and item embeddings on item side (LIME-MLP)
> * XOR-style attention on the user side and MLP on the item side (LIME-XOR-MLP Hybrid)
>
> These variants performed significantly worse than LIME-XOR and clearly LIME-XOR’s design on the user side is especially crucial to strong performance. This is because LIME-XOR adeptly handles variable length sequences and gives a low-rank approximation to self-attention.
>
> **(a) Ablation Study**
>
> | Model Variant               |   AUC      |
> |-----------------------------|------------|
> | LIME-XOR                    | **0.7448** |
> | LIME-XOR w/ Link Pooling    |  0.7400    |
> | LIME-XOR w/ Dot Product     |  0.7404    |
> | LIME-MLP                    |  0.7407    |
> | LIME-XOR–MLP Hybrid         |  0.7442    |
>
>
>
> Regarding highly optimized transformer variants, our benchmark comparisons are now all based on Flash Attention v2 implemented in Triton (explained more in the question above). We also added several popular linear attention variants to the comparison (see table 6(b) in Section 4.8) which we reproduce below. In terms of both complexity and AUC, LIME-XOR performs better than other popular linear attention alternatives since our low-rank $\ell \ll N(U), M(U)$.
>
> **(b) Linear Attention Comparison**
>
> | Model Variant        |   AUC     | Complexity                           |
> |----------------------|-----------|--------------------------------------|
> | HSTU Skyline         |  0.7444   | O(N(U)² + N(U) · M(U))               |
> | Linear Transformer   |  0.7423   | O(N(U) · M(U))                       |
> | Longformer           |  0.7443   | O(N(U) · M(U))                       |
> | LREA (Linformer)     |  0.7408   | O(N(U) · M(U))                       |
> | LIME-XOR             |  0.7448   | O(N(U)ℓ + M(U)ℓ)                     |
> | LIME-XOR+Window      | **0.7453**| O(N(U)ℓ + M(U)ℓ)                     |
>
>
> We also introduce LIME-XOR+Window, which augments LIME-XOR with a local sliding-window mask of size $\ell$ to capture additional local interactions. This yields improved results on both KuaiRand and our industrial dataset, while keeping complexity comparable to LIME-XOR. Since any form of linear attention can be integrated into our decoupled transformer LIME framework, in the future we can explore other sophisticated techniques such as Mamba [4], MOBA [5], and Kimi Linear [6].
>
> [4] Gu, Albert, and Tri Dao. Mamba: Linear-Time Sequence Modeling with Selective State Spaces.
>
> [5] Pióro, Maciej, Kamil Ciebiera, Krystian Król, Jan Ludziejewski, Michał Krutul, Jakub Krajewski, Szymon Antoniak, Piotr Miłoś, Marek Cygan, and Sebastian Jaszczur. MoE-Mamba: Efficient Selective State Space Models with Mixture of Experts.
>
> [6] K Team. Kimi Linear: An Expressive, Efficient Attention Architecture.

---

> ### Author Response · Authors · 2025-11-21
>
> > *It is of great importance to add comparisons with more recently developed accelerated attention mechanisms. These comparisons should evaluate both accuracy and efficiency across a comprehensive range of hyperparameters, including user sequence length, number of candidates, and QK dimensionality. The comparisons in the existing literature should also be unified to include these aspects.*
>
> The Hierarchical Sequential Transducer Unit (HSTU) is a state-of-the-art transformer variant for RecSys that applies causal self-attention over user history and target attention from candidates, fused in a single kernel via a trapezoidal mask. Comparisons with standard transformers are detailed in the HSTU paper [7], and we also include SASRec [8], which applies a transformer over the full user history and candidate sequence without separate target attention.
>
> Table 4 (Appendix F) reports exhaustive parameter sweeps of several baselines along with LIME across sequence length, candidate count, and QK dimension. Figures 9–10 (Appendix J) compare LIME-XOR to other linear attention variants. In terms of both accuracy and efficiency across a range of hyperparameters, LIME consistently outperforms other linear attention approaches.
>
> [7] Actions Speak Louder than Words: Trillion-Parameter Sequential Transducers for Generative Recommendations — Zhai et al.
>
> [8] SASRec: Self-Attentive Sequential Recommendation — Kang et al.
>
> > *The manuscript should include a discussion section addressing the limitations of the proposed approach and outlining potential directions for future research to address these shortcomings.*
>
> Thanks for the suggestion. We have added the requested section 6. The main future direction we’d like to pursue is a sparsified version of the link embeddings to account for more fragmented user interest. For limitations, we also highlighted that LIME is best suited for early stage ranking where the number of candidates to score is typically in the 10s of thousands, much larger than late stage ranking. We would also like to expand our work to further benefit cold-start users (of which there is some empirical evidence we added in Appendix L).
>
> > *In comparison with the baseline MSA, the authors should specify the details of its implementation. It is unclear whether MSA is implemented using the modern FlashAttention mechanism (which could be essential for fairness). The authors are advised to clarify this point and, if FlashAttention is used, to indicate the specific version employed.*
>
> Thank you for the reminder. To ensure LIME is competitive for online productionization, we have implemented Triton/FlashAttention v2- based kernels for both the multi-head attention and the XOR attention. While our initial benchmarks used native PyTorch simply to demonstrate LIME's scalability on long sequences, we have now updated the skyline models (MHA and HSTU skyline) in the latest paper revision to also use Triton, specifically with FlashAttention v2 for the attention mechanism. Furthermore, we have added the derivation of the forward and backward algorithms for the XOR attention kernel (which uses SiLU activation) to Appendices H and I.

---

### Official Review · Reviewer_619M · 2025-10-30

**Soundness:** 2
**Presentation:** 3
**Contribution:** 2
**Rating:** 6
**Confidence:** 3

**Summary:**

LIME addresses the efficiency-expressiveness trade-off in large-scale recommendation systems via two innovations: low-rank link embeddings and XOR attention (LIMEXOR). Link embeddings decouple user-item interactions, enabling offline pre-computation of attention weights to make inference latency nearly independent of candidate set size. LIMEXOR reduces self-attention complexity from quadratic to linear by restricting interactions between user history and link embeddings. Experiments on public and industrial datasets show LIME matches the accuracy of expressive models (e.g., HSTU) while achieving 10× lower inference latency, with online A/B tests yielding up to 37.9% gains in user engagement.

**Strengths:**

- The paper simultaneously resolves two bottlenecks—candidate set scaling (via pre-computed attention weights) and long user history modeling (via linear XOR attention).
- Near-SOTA Accuracy with Low Latency: LIME-XOR closes the performance gap with HSTU across tasks (video completion, watch time) while maintaining latency that is orders of magnitude lower.
- Practical Deployment Value: Its decoupled inference pipeline (offline item pre-computation + online user processing) aligns with real-world recommendation systems, and online A/B tests confirm tangible user engage

**Weaknesses:**

- Link Count Sensitivity: Performance scales with link count, but small (e.g., 8–16) underperforms, and large may offset efficiency gains
- Limited Shorter Sequence Advantage: On datasets with short user histories (e.g., Taobao Ads, max length 50), LIME-XOR performs only slightly better than LIME-MHA and lags HSTU. It seems that its linear attention gains are less impactful for small N.
- Cold-Start Unaddressed: The framework relies on historical user interactions to personalize link embeddings, rarely consider cold-start users/items.

**Questions:**

How does LIME perform when link embeddings are initialized with task-specific priors (e.g., item category clusters) instead of random normal distributions?

Could a dynamic link count (adjusted per user’s history length) balance efficiency and performance, and what mechanisms would prevent overfitting to short sequences?

---

> ### Author Response · Authors · 2025-11-22
>
> We thank the reviewer for careful review, accurate summary, and thoughtful feedback. We made changes to the paper based on your suggestions, highlighted in blue. Below we address each of your concerns
>
> > *Link Count Sensitivity: Performance scales with link count, but small (e.g., 8–16) underperforms, and large may offset efficiency gains*
>
> While performance does drop when using very small link counts such as 8 or 16, the degradation is modest compared to the 32-link setting. In highly capacity-constrained inference scenarios, even 8 links achieve strong performance on our industrial dataset, with only a ~0.1% NE (normalized entropy) regression relative to 32 links. Moreover, as sequence length grows, scaling the number of links continues to yield consistent performance gains (see Appendix G, Table 5 for our scaling-law study).
>
> | Number of Links | AUC on KuaiRand-1k | VC NE Gain on Industrial Dataset |
> |:--------------:|:------------------:|:-------------------------------:|
> | 2              | 0.7416             | -0.81%                          |
> | 4              | 0.7431             | -0.89%                          |
> | 8              | 0.7440             | -0.96%                          |
> | 16             | 0.7448¹            | -1.01%                          |
> | 32             | 0.7449             | -1.04%²                         |
> | 64             | 0.7445             | -1.08%                          |
> | 128            | 0.7447             | -1.24%³                         |
> | 256            | 0.7451             | -1.25%                          |
>
> Footnotes:
>
> 1. Performance saturates, reported in Figure 5(b).
>
> 2. Reported in Figure 5(a).
>
> 3. Performance saturates.
>
>
> In most practical deployments, allocating 32 or more links is easily feasible—and even this represents a substantial compression (e.g., a 32× reduction for a user history of length 1024 in our industrial dataset). This efficiency is significant because user histories commonly span thousands to tens of thousands of events, and future applications may reach hundreds of thousands. Such scales are typical in short-form video platforms, as noted in Twin V2 [2], LONGER [3], and PinFM [4]. Our setting is even broader, as we incorporate both video and non-video interactions, as well as cross-surface user behaviors.
>
> [2] TWIN V2: Scaling Ultra-Long User Behavior Sequence Modeling for Enhanced CTR Prediction at Kuaishou
>
> [3] LONGER: Scaling Up Long Sequence Modeling in Industrial Recommenders
>
> [4] PinFM: Foundation Model for User Activity Sequences at a Billion-scale Visual Discovery Platform
>
> > *Limited Shorter Sequence Advantage: On datasets with short user histories (e.g., Taobao Ads, max length 50), LIME-XOR performs only slightly better than LIME-MHA and lags HSTU. It seems that its linear attention gains are less impactful for small N.*
>
> The benefits of LIME are most pronounced for longer user histories (see Appendix G, Table 5), aligning with the broader industry trend toward modeling increasingly long sequences, as seen in Twin V2, LONGER, and PinFM. Inspired by analogous challenges in LLMs, LIME is designed to efficiently encode a user’s full interaction history and understand the evolution of user interests over time.
>
> To assess performance on short histories, we evaluated pretrained HSTU Skyline and LIME-XOR models on truncated sequences in KuaiRand-1K (see Appendix L). Across all short-sequence thresholds, LIME-XOR consistently outperforms HSTU. Moreover, our industrial A/B test shows that LIME also delivers topline gains for cold-start users, matching the improvements observed in the overall cohort. One explanation is that LIME compresses each user history into a fixed-length embedding, providing a normalization effect that improves robustness. In contrast, Transformer models like HSTU may be more sensitive to variations in history length due to direct cross attention from item to user history.
>
> | Cold-Start User Sequence Length Threshold | LIME-XOR | HSTU   |
> |-------------------------------------------|----------|--------|
> | 4                                         | **0.7022**   | 0.7016 |
> | 8                                         | **0.7179**   | 0.7173 |
> | 16                                        | **0.7284**   | 0.7273 |
> | 32                                        | **0.7369**   | 0.7356 |
>
> Given the strong cold-start performance of LIME-XOR on KuaiRand and the online A/B test, shorter sequences alone may not explain why LIME-XOR underperforms HSTU on the Taobao dataset. In general, LIME was initially designed for longer sequence modeling but we agree the important problem of cold-start user performance requires a dedicated and careful study.

---

> ### Author Response · Authors · 2025-11-22
>
> > *Cold-Start Unaddressed: The framework relies on historical user interactions to personalize link embeddings, rarely considers cold-start users/items.*
>
> We thank the reviewer for raising this concern and agree that cold-start users and items are an important challenge. As noted above, LIME delivers improved cold-start performance on both KuaiRand and our online A/B tests. In our industrial dataset, we further leverage user contextual features (see link contextualization in Figure 3), such as demographic attributes, to provide a strong global prior when explicit history is unavailable.
>
> For cold-start items, one popular strategy is to rely on content-based signals (e.g., video pixels or captions) to produce meaningful item token embeddings. In this work, we use item metadata (e.g., topic, length, popularity) when generating token embeddings, which prevents the model from relying solely on ID-based embeddings (see Model Overview on page 3). Looking ahead, we plan to incorporate richer multimodal signals into the token and link embeddings, as outlined in our newly added Section 6. These enhancements integrate naturally into LIME’s decoupled attention framework, since token embeddings are fully flexible.
>
> > *How does LIME perform when link embeddings are initialized with task-specific priors (e.g., item category clusters) instead of random normal distributions?*
>
> We appreciate the suggestion. This direction aligns closely with our ongoing work on hard links, where links are explicitly associated with item topic clusters (in contrast to LIME’s current soft-link formulation), as noted in Section 6. Because topic clusters have much larger cardinality, we will need mechanisms to sparsify link assignments—e.g., selecting only the top-k relevant links per user—which we expect will improve both scalability (analogous to MoE routing) and interpretability.
>
> Interestingly, our analysis already reveals that soft links exhibit emergent clustering behavior. Items receiving the highest attention weights for a given soft link show clear semantic similarity: some links correspond to video length, others to content topics (e.g., food, movies), and others to language (see Appendix K). For privacy reasons we provide only high-level category descriptions, though the clustering is even more apparent when examining the actual thumbnails. This suggests that soft links naturally discover meaningful item groups, focusing on the most salient categories given the limitation on the number of links. We believe that combining soft and hard links in the future may give the strongest overall performance.
>
> > *Could a dynamic link count (adjusted per user’s history length) balance efficiency and performance, and what mechanisms would prevent overfitting to short sequences?*
>
> This is a very natural suggestion. Currently for simplicity in productionization we used a fixed number of links per user, which is also better aligned with hardware allocation during serving with cached QK attention weights. But making the number of links dynamic based on user interest diversity is a great idea to further improve storage ROI. To prevent overfitting, we have used normalization (mainly RMSNorm) extensively in the LIME architecture. For cold-start users with short sequences, we observe strong gains in both our online A/B test and the KuaiRand evaluation (Appendix L, Table 7), suggesting no evident overfitting in practice.

---

### Official Review · Reviewer_ZWMY · 2025-10-31

**Soundness:** 3
**Presentation:** 4
**Contribution:** 3
**Rating:** 8
**Confidence:** 3

**Summary:**

This paper presents LIME, a novel recommendation architecture aimed at improving efficiency while maintaining performance. The core contribution lies in decoupling the internal structure of attention and restructuring modules through two key components: LIME-MHA and LIME-XOR. LIME-MHA precomputes and caches candidate–link attention weights offline, effectively replacing the traditional cross-attention process with a matrix multiplication between precomputed item-side link matrices and personalized link embeddings. LIME-XOR further enhances efficiency by reducing self-attention complexity from quadratic to linear via an XOR masking scheme for user–link interactions. Extensive experiments on public and industrial datasets, including online A/B tests, confirm significant efficiency improvements and competitive or superior accuracy compared to transformer baselines.

**Strengths:**

- Novel architectural design that effectively resolves the efficiency–expressiveness trade-off in large-scale recommendation.
- Clear technical innovations in decoupling attention (LIME-MHA) and introducing efficient XOR masking (LIME-XOR).
- Detailed experimental verification across public and industrial datasets, with both offline and online A/B test results provided.
- Demonstrated scaling properties and robustness in latency under very large candidate sets and long user histories.

**Weaknesses:**

If I haven't misunderstood, the LIME-MHA part actually replaces the attention mechanism with a well designed matrix multiplication between user and candidate parts, manifested as the matrix multiplication of one matrix TL^t (independent of the user's item-side information) and the second one L^P (user-side personalized information). Therefore, this approach will definitely significantly enhance the speed of reasoning. However, based on the results of this article, its performance is also very strong. Does this prove that the personalized matrix multiplication is equally or even more effective than MHA in some recommendation aspects? This is a question worth exploring.

**Questions:**

If I haven't misunderstood, the LIME-MHA part actually replaces the attention mechanism with a well designed matrix multiplication between user and candidate parts, manifested as the matrix multiplication of one matrix TL^t (independent of the user's item-side information) and the second one L^P (user-side personalized information). Therefore, this approach will definitely significantly enhance the speed of reasoning. However, based on the results of this article, its performance is also very strong. Does this prove that the personalized matrix multiplication is equally or even more effective than MHA in some recommendation aspects? This is a question worth exploring.

---

> ### Author Response · Authors · 2025-11-22
>
> We thank the reviewer for thoughtful review and summary. We made changes to the paper based on your suggestions, highlighted in blue. We address your questions below.
>
> > *If I haven't misunderstood, the LIME-MHA part actually replaces the attention mechanism with a well designed matrix multiplication between user and candidate parts, manifested as the matrix multiplication of one matrix TL^t (independent of the user's item-side information) and the second one L^tP (user-side personalized information). Therefore, this approach will definitely significantly enhance the speed of reasoning. However, based on the results of this article, its performance is also very strong. Does this prove that the personalized matrix multiplication is equally or even more effective than MHA in some recommendation aspects? This is a question worth exploring.*
>
> LIME compresses user history and context (e.g. demographic info) into a small set of link embeddings, enabling candidate items to attend to these rather than the full history. This design improves both efficiency and accuracy.
>
> To address your question, we conducted (see Figure 6) the following experiments that adhere to a more straightforward personalized matrix multiplication approach instead of our decoupled attention approach
>
> * LIME-MLP:  we pad user history embeddings to fixed length, and feed them to an MLP to generate a low-rank user history summary. Then, on the item side we generate weights with an MLP to weighted pool the low-rank summary.
>
> * Linear Transformer [1]: where we replace $\phi(QK^t) V$ with $\phi(Q) (\phi(K)^t V)$ where $\phi(K)^t V)$ is computed first. This can be viewed as the product of two matrices: $\phi(K)^t V$ which is a $d \times d$ matrix purely from the user side and $\phi(Q)$ which is a $m \times d$ matrix from the candidate items.
>
> | Model             | AUC (KuaiRand-1k) |
> |-------------------|-------------------|
> | LIME-XOR          | **0.7448**            |
> | LIME-MLP          | 0.7407            |
> | Linear Transformer| 0.7423            |
>
> These alternatives consistently underperform LIME (more variants evaluated in the ablation study in Figure 6(a)), highlighting that our specific decoupled attention design is essential for achieving both high efficiency and strong recommendation accuracy.
>
> Regarding LIME-XOR vs HSTU, on both industrial datasets and public datasets we indeed found that the link compression from LIME-XOR causes very little regression. However as the user sequence length increases, we do need to increase the number of links slightly as demonstrated in Table 5 of appendix G to accommodate the increasing amount of personalization capacity.
>
> [1] Transformers are RNNs: Fast Autoregressive Transformers with Linear Attention (ICML 2020)

---

### Author Response · Authors · 2025-12-01
**Rebuttal Summary to AC**

Dear Area Chair,

We are sorry to hear about the information leak during the review process and thank the AC for taking the time to review our submission. For your convenience, we summarize the key contributions and strengths outlined by reviewers and our rebuttal to their reviews.

**Our primary contribution is a novel decoupled framework for multi-layer transformers in recommendation systems** enabling highly performant inference at a massive scale where both user history lengths and candidate sets scale to tens of thousands of items. We do this by:
* Introducing link embedding sequences to summarize a user’s history
* Developing a QK-cache mechanism to precompute attention weights offline
* Proposing a novel XOR attention mask to factorize self-attention
* Demonstrating on-par accuracy with expensive Transformer models such as HSTU with **10x lower latency** and deploying LIME in production with **38% source rate gain** in online A/B tests

These novelties were acknowledged by all three reviewers with the following highlights:
* Novel architecture resolving efficiency-expressiveness tradeoffs in large-scale recommendation systems (ZWMY, 619M)
* Strong accuracy with lower latency when scaling user histories and candidate sets (ZWMY, 619M, VtZ9)
* Practical deployment and detailed experimental verification with both offline results and online A/B test (ZWMY, 619M, VtZ9)

We also would like to summarize our rebuttal responses below (changes to our paper are highlighted in blue).

To address reviewer ZWMY’s concerns:
* We experiment with several alternate variants to design a decoupled personalized matrix multiplication approach through either linear transformers or replacing one of the two attention stages with MLP. However, we observe a clear performance drop highlighting the effectiveness of the LIME architecture.

To address reviewer 619M’s concerns:
* We performed additional experiments on a wide range of link counts. We see competitive performance even with few links (32x compression rate of the user history is reported in Figure 5(a)).
We provided slice metrics on short user sequences (cold-start users), which still showed significant gains in both public and industrial datasets.
* We provided semantic clustering evidence in Appendix K that the links represent meaningful interest topicality.
* On the comparison between LIME and HSTU, we highlighted that since LIME scores a much larger number of candidates (10s of thousands as opposed to hundreds), we simply cannot afford a full self-attention + cross-attention architecture like HSTU.
   * Critically, our production deployment of LIME brings **+37.9% VC** (video completion task) from ranked candidates in online A/B tests and on both public & industrial datasets LIME has on-par performance with HSTU.

To address reviewer VtZ9’s concerns:
* We reran all the benchmarks with Triton implemented attention kernels based on FlashAttention V2 algorithm, and reached the same scaling law conclusion as PyTorch. Detailed kernel pseudo code and gradient derivation are also attached in the appendices.
   * Note that our latency benchmarks do not fully account for LIME’s decoupled design, which allows the multi-layer Transformer user tower to be precomputed offline or executed at the start of a request in parallel with candidate retrieval. With this latency masking, LIME’s true production latency savings are far greater—**over 90%**—since the only exposed cost is a very lightweight weighted sum.
* We ran a systematic ablation study of the various components in LIME-XOR, including replacing link personalization component with sum pooling, replacing either of the two attention stages with MLP or dot product, as well as substituting standard linear attention variants for the XOR-attention across a range of hyperparameters. We observe all ablated variants and linear attention alternatives lead to a steep drop in performance.
   * Thanks to the flexibility of LIME’s decoupled architecture, we can also plugin other linear attention techniques into the user module. We experimented with incorporating sliding window attention into LIME-XOR which further improved accuracy leading to our advanced variant LIME-XOR+Window.
* Similar to reviewer 619M, we measured LIME’s utility for shorter sequences through slice metrics. But the main focus of LIME is to handle user sequences in the range of 10s or 100s of thousands, which are increasingly common in industrial scale recommendation systems.

Finally we added a section on limitations and future directions, highlighting the unique suitability of LIME in early stage ranking components, which are relatively under-explored.

We believe the reviewers were initially positive, and although they could not respond—our rebuttals were posted on 11/21, six days before the leak incident—we conducted additional experiments to address their questions. We thank the reviewers for helping strengthen our submission.

Best,

LIME Authors

---

### Note · Program_Chairs · 2026-01-17
**Submission Desk Rejected by Program Chairs**

The following references in this submission do not refer to real documents and/or have major errors in bibliographic information:

 Pan Li, Yuting Su, Xiao Sun, et al., “Eta: A tmall-ads framework for efficient user behavior modeling with full-interaction,” Proceedings of the 29th ACM SIGKDD Conference on Knowledge Discovery and Data Mining (KDD), pp. 4307–4316, 2023.